# Physical Disorders are Associated with Health Risk Behaviors in Chinese Adolescents: A Latent Class Analysis

**DOI:** 10.3390/ijerph17062139

**Published:** 2020-03-23

**Authors:** Bingdong Song, Weirong Hu, Wanxia Hu, Rong Yang, Danlin Li, Chunyu Guo, Zhengmei Xia, Jie Hu, Fangbiao Tao, Jun Fang, Shichen Zhang

**Affiliations:** 1Department of Toxicology, School of Public Health, Anhui Medical University, and Key Laboratory of Environmental Toxicology of Anhui Higher Education Institutes, No 81 Meishan Road, Hefei 230032, China; bingdongsong@163.com (B.S.); 13805594603@163.com (W.H.); yangrong@stu.ahmu.edu.cn (Y.R.); lidanlin@stu.ahmu.edu.cn (D.L.); ChunyuGuo1029@163.com (C.G.); xiazhengmeiah@163.com (Z.X.); hujie@stu.ahmu.edu.cn (J.H.); fbtao@ahmu.edu.cn (T.F.); 2Department of Maternal, Child and Adolescent Health, School of Public Health, Anhui Medical University, and MOE Key Laboratory of Population Health Across Life Cycle/Anhui Provincial Key Laboratory of Population Health and Aristogenics, No 81 Meishan Road, Hefei 230032, Anhui, China; 3School of Health Management, Anhui Medical University, No.81, MeiShan Road, Hefei 230032, Anhui, China; wanxiahu@stu.ahmu.edu.cn; 4Faculty of Pharmaceutical Science, Sojo University, Ikeda 4-22-1, Kumamoto 860-0082, Japan

**Keywords:** health risk behaviors, physical disorders, adolescents, latent class analysis

## Abstract

It is known that health risk behaviors (HRBs) can lead to a variety of physical and mental health problems among adolescents, but few studies have paid attention to the relationship between latent classes of HRBs and adolescent diseases. The purpose of this study was to use latent class analysis (LCA) to clarify the potential subgroups of HRBs (smoking, drinking, screen time, non-suicidal self-injuries, suicidal behaviors, and unintentional injuries) and examine the association between the subgroups of HRBs and physical disorders (diarrhea, fever, cough, and vomiting) with multiple logistic regression analysis, in Chinese adolescents. Self-reported HRBs and physical disorders were used to evaluate 22,628 middle school students in six cities of China, from November 2015 to January 2016, based on a multistage stratified cluster sampling approach. The prevalence of diarrhea, fever, cough, and vomiting was 23.5%, 15.9%, 50.6%, and 10.7%, respectively. We identified four latent classes of HRBs by LCA, including low-risk class, moderate-risk class 1 (smoking, drinking, and screen time), moderate-risk class 2 (non-suicidal self-injuries and suicidal behaviors, unintentional injuries), and high-risk class (smoking, drinking, screen time, non-suicidal self-injuries, suicidal behaviors, and unintentional injuries), which were 64.0%, 4.5%, 28.8% and 2.7% of participants, respectively. Compared to the low-risk class, all other classes showed higher risk for these physical disorders (*P* < 0.01 for each). In particular, the high-risk class had the highest risk (diarrhea (odds ratio (OR) = 2.628, 95% confidence interval (CI) 2.219 to 3.113), fever (OR = 3.103, 95% CI 2.591 to 3.717), cough (OR = 2.142, 95% CI 1.805 to 2.541), and vomiting (OR = 3.738, 95% CI 3.081 to 4.536). In conclusion, these results indicated that heterogeneity exists in HRBs, and subgroups of HRBs were correlated to the occurrence of common physical disorders in Chinese adolescents. Therefore, multiple HRBs rather than single factors should be considered for the prevention of common physical disorders in schools.

## 1. Introduction

The health of adolescents has been becoming a critically important problem which is, however, always neglected and the health risk behaviors (HRBs) of adolescents has greatly increased the burden of many diseases or physical disorders and resulted in social problems [1,2]. The physical disorders commonly seen in adolescents include diarrhea, fever, cough, and vomiting [3]. The Global Burden of Disease study reported that, from 1980 to 2016, diarrhea was the eighth leading cause of death worldwide, with more than 16.5 million deaths every year [4]. One third of adolescents suffer from diarrhea in developing countries [5] and a study found that fever and diarrhea often appear together in patients [6]. Fever was the most common symptom of health care needs in the tropics and low-income countries. Fever can occur alone, but it can also appear together with a cough [7]. It is worth noting that some studies have reported that acute fever can cause death [8]. In addition, cough was a common clinical symptom, which was related to pneumonia, bronchial asthma, infection, etc., threatening the health of children and adults [9,10]. Without suitable and effective treatment, a continuous long-term cough (e.g., > eight weeks) can become a chronic cough [10]. The global prevalence of chronic cough is 9.6%, and in Europe and the United States, the prevalence of chronic cough is 12.7% and 11.0%, respectively, whereas the prevalence of chronic cough is 4.4% in Asia [11]. In addition, the incidence of vomiting which can occur in many cases including drug therapy [12], drinking, etc. [13], should also not be ignored.

Previous studies have shown that HRBs are closely related to diarrhea, fever, cough, and vomiting [14]. A study in the general population by Çolak, Y. found that the prevalence of cough was 3% among non-smokers, 4% in former smokers, while it increased to 8% in current smokers. There was a positive correlation between smoking and cough prevalence [15]. Moreover, a study in Japan showed that drinking was significantly associated with vomiting. The incidence of vomiting in non-drinkers was 11% and it increased to 30% in alcoholics [16]. The alcohol-related consequences have also been confirmed in other studies [17]. It should be noted that the injury caused by smoking and drinking is cumulative and that smoking and drinking at a young age can cause more serious health injury than starting these behaviors in adult, such as high incidence of cardiovascular disease, chronic respiratory disease, and even cancer [18]. In addition, a study found that 43% of adolescents, in the United States, have excessively long screen time, and excessive screen time results in a lower quality of life [19]. Furthermore, a separate study found that poor quality of life is significantly associated with cough [20]. Hagiya, K. reported repeated severe vomiting and diarrhea when a nurse attempted suicide by taking intravenous nicotine [21]. In a survey study about non-suicidal self-injuries, Svirko, E. found that there was a close relationship between eating disorders and non-suicidal self-injuries. Patients with eating disorders usually experience non-suicidal self-injuries and vice versa [22]. Related research has also confirmed that eating disorders can cause gastrointestinal diseases, diarrhea, and vomiting [23,24]. More importantly, there is a significant correlation between non-suicidal self-injuries and suicidal behaviors, and those with non-suicidal self-injuries usually have higher suicidal behaviors [25]. Unintentional injuries have been divided into many types (such as road traffic injuries, poisoning, falling or tripping injuries, etc.), which can cause a variety of adverse results, for example, an outbreak of acute gastroenteritis at a university in northwest China caused diarrhea, fever, and vomiting in most students [26]. Trommel, N. and others conducted a 30-year retrospective study and found that burns can cause fever and vomiting [27]. Singh, A.K.’s study also found that vomiting is one of the main symptoms of road traffic injuries [28].

These results suggest a clear association between HRBs and common physical disorders. However, many current studies focused on the relationship between a single risk factor and physical disorders but did not investigate the interactions of different HRBs and the combined effects of different HRBs. It should be noted that HRBs usually do not occur alone, but often appear simultaneously. In other words, traditional variable-centered research (such as factor analysis and regression analysis) does not provide critical information on the intersection of various HRBs [29], therefore, we need to use an innovative statistical method, which can explore how HRBs cluster among individuals, and therefore deepen people’s understanding [30]. In summary, current research proposes the following hypothesis: (1) potential subgroups of HRBs could be observed and (2) subgroups of HRBs are associated with physical disorders.

## 2. Methods

### 2.1. Study Design: Participants and Procedures

This study was a cross-sectional study conducted in six different cities in China after approval by the Ethics Committee of the Anhui Medical University (1 March 2014, approval number 20140087). This study was carried out from November 2015 to January 2016. For all adolescents enrolled in this study, informed consents from their parents were requested and obtained. This study used a multistage stratified cluster sampling approach to recruit participants from junior and senior high schools in six Chinese cities which were Shenyang (capital of Liaoning Province), Xinxiang (North of Henan Province), Yangjiang (Southwest coast of Guangdong Province), Chongqing (one of China’s four direct-controlled municipalities), Ulanchap (Central Inner Mongolia Autonomous Region), and Bengbu (Northeastern of Anhui province), including urban and rural areas. Specifically, the six cities were selected by convenient sampling, and then eight schools were selected in each region by stratified cluster sampling (two middle schools and two high schools in rural areas (the numbers were 5186 and 5696, respectively), two middle schools and two high schools in the urban areas (the numbers were 6807 and 4939, respectively)). Finally, four to six classes of students were randomly selected from each grade (grades seven to twelve) in each school, all students from the selected classes were invited to the study.

Research staff explained the purpose and process of the study to the students participating in the survey, and students were allowed to choose to participate or not. Each respondent independently completed a self-reported questionnaire that included sociodemographic characteristics, smoking, alcohol consumption, non-suicidal self-injuries, suicidal behaviors, unintentional injuries, screen time, and occurrence of physical disorders (i.e., diarrhea, fever, cough, and vomiting) during two week before the survey study. After excluding invalid questionnaires with missing data (>5%), a total of 22,628 valid questionnaires were obtained, with an effective rate of 97.8%. 

### 2.2. Characteristics of the Sample

The average age of the 22,628 respondents in this study was 15.4 ± 1.8 years, comprised of 10,990 boys (48.6%) and 11,638 girls (51.4%). The respondents included 11,993 middle school students (53%), and 10,635 high school students (47.0%); 11,802 students (48.1%) in rural areas, 11,746 (51.9%) students in urban areas; 9720 (43.0%) students with one child in the family, 12,908 (57.0%) students with more than one child in the family; 11,320 students (50.0%) who were boarding on school days, 11,308 students (50.0%) who were commuting student; 13,006 (57.5%) students whose fathers were below the high school degree; 9424 (41.6%) students whose fathers were above the high school degree; 14,335 (63.4%) students whose mothers were below the high school degree, 8105 (35.8%) students whose mothers were above the high school degree; 3240 (14.3%), 16,345 (72.2%) and 3043 (13.4%) students who self-reported family economy with bad, general, and good, respectively; and 5514 (24.4%), 9620 (42.5%), and 7494 (33.1%) students with 2 or fewer friends, 3 to 5 friends, and 6 or more friends, respectively.

### 2.3. Questionnaire and Measures

To evaluate the physical disorders, the students were asked if the following statuses occurred in the past 2 weeks: diarrhea (at least 3 times in 1 day or 24 hours), fever (body temperature above 37.5 °C), cough (one or more times), and vomiting (one or more times). The answer options include “0 = 0 time and 1 = one time and above”. The validity and reliability of the two-week prevalence rate have been demonstrated in [31].

According to the definition by the Centers for Disease Control and Prevention of Youth Risk Behavior Surveillance System [32], current smoking and drinking status was measured using the following questions: How many times have you smoked in the past 30 days? and How many days have you had at least one glass of wine in the last 30 days? The options for answering the two questions were “0 = 0 day, 1 = 0 to 9 days, 2 = 10 to 19 days, 3 = 20 to 30 days”. For these two items, choose 0 as no and other options as yes [32]. The validity and reliability of smoking and drinking were verified by a previous study [33].

Screen time was assessed using questions based on the youth risk behavior questionnaire [34]. Questions were asked about the average time per day spent on games, music players, tablets, smartphones, QQ, Weibo, Renren, and other social tools, as well as on the Internet, or doing somethings unrelated to learning on the computer during school days. Screen time >2 h was defined as too much screen time [35]. There were 3690 adolescents with screen time over 2 h among all the research subjects, accounting for 16.3%.

Regarding non-suicidal self-injuries, a self-report questionnaire was used to assess non-suicidal self-injuries of students over the previous 12 months [36] which included the following eight aspects: (1) beating yourself deliberately, (2) pulling your hair deliberately, (3) hitting your head or hitting other objects with your fist deliberately, (4) swearing yourself or scratching yourself deliberately, (5) biting yourself deliberately; (6) cutting or stabbing oneself deliberately, (7) deliberately overdosing with drugs, alcohol, or smoking, etc., (8) deliberately swallowing foreign objects (such as indigestible items). All the response options were “yes” or “no”. If the answer was “yes” (one or more times), the student was defined as having non-suicidal self-injuries. The Cronbach’s α coefficient for this study was 0.779. 

According to the Youth Risk Behavior Surveillance System, suicidal behaviors were composed of suicide ideation, suicide planning, and suicide behavior [32]. In this study, suicidal ideation, suicide planning, and suicide behavior were investigated by the following questions: “Had you seriously thought about taking your own life in the past year?”, “Had you seriously thought about or written about suicide in the past year?”, and “Had you tried suicide in the past year?”, respectively. Four options were provided for each question, which were “1 = none, 2 = 1 time, 3 = 2 to 3 times, 4 ≥ 4 times”. The students who had one or more of the above behaviors were identified as having suicidal behaviors [33]. The validity and reliability of suicidal behaviors were demonstrated [33]. 

Unintentional injuries were assessed by the National Standards of the People’s Republic of China (a method for injury surveillance among children and adolescent) [37], which consisted of 12 classes (road traffic incident, crush, falling and tripping, scratches, puncture or cut, bites and pricks, explosive impact, enclosed anoxic space, drowning, electric shock, chemical or other substances poisoning, and other injuries). The participants were asked if one or more of the above classes of injuries had occurred in the past year, and those who answered “yes” were considered to have suffered unintentional injuries. The Cronbach’s α coefficient for this study was 0.724. 

### 2.4. Statistical Analysis

The database was created with EpiData 3.1 software, and statistical analyses were performed with SPSS 23.0 and Mplus 7.4. Moreover, SPSS was used for the normality test, Chi-square test, and confidence interval test. The Chi-square test was applied to compare the incidence of physical disorders among different demographic variables. Multivariate logistic regression models were used to examine the associations of HRBs and physical disorders. Odds ratios (OR) and 95% confidence intervals (CI) for the factors were calculated to present the associations. In the analyses, *p* < 0.05 was considered statistically significant.

Latent class analysis (LCA) procedures were used to identify subgroups (classes) of respondents who exhibited distinct constellations of substance use, thus, capturing heterogeneity associated with unobserved or latent subpopulations. LCAs were conducted using Mplus (Version 7.4), with maximum likelihood estimation. The best fitting class was selected based on the following criteria: (1) Akaike information criterion (AIC) in which the smaller AIC value indicates a better fit model; (2) Bayesian information criterion (BIC) in which the smaller BIC value indicates a better fit model; (3) adjusted Bayesian information criterion (aBIC) in which the smaller aBIC value indicates a better fit model; (4) Lo–Mendell–Rubin (LMR) likelihood ratio in which lower and significant *P*-values (*P* < 0.05) indicate that the evaluation model is better than others; (5) bootstrapped likelihood ratio test (BLRT) in which significant *P*-values (*P* < 0.05) indicate a better fit model; and (6) entropy based on the posterior class membership probabilities which indicates the extent to which each class is separated and represented by data, in which the closer the value of entropy is to “1”, the clearer the model separation [38,39,40].

## 3. Results

### 3.1. Demographic Characteristics 

The prevalence of diarrhea, fever, cough, and vomiting was 23.5%, 15.9%, 50.6%, and 10.7%, respectively. The frequency characteristics are shown in Table 1. Significant differences of all physical disorders (diarrhea, fever, cough, and vomiting) were found in grades (*P* < 0.01 for each). In addition, the students of high school, rural, boarding on school days, bad self-reported family economy, and less friends reported higher rates of diarrhea than the correspondence groups (*P* < 0.01 for each). Fever was more common in students of middle school, male, rural, non-boarding, high educational level of mother, good self-reported family economy, and less friends (*P* < 0.05 for each). Cough was reported more in students of middle school, female, and no siblings (*P* < 0.01 for each). Vomiting was found more in students of middle school, rural, boarding on school days, good self-reported family economy, and less friends than the correspondence groups (*P* < 0.01 for each, Table 1).

### 3.2. Latent Class Analysis of HRBs

We determined the pattern of HRBs based on LCA. The five-classes model did not replicate the best log likelihood values and LMR-LRT was not statistically significant (*P* > 0.05), and therefore was not considered further. Then, according to the series of indicators, as shown in Table 2, the four latent classes were considered to be the best models because they had the lowest AIC, BIC, and aBIC values. In addition, LMR-LRT and BLRT were statistically significant in the four-classes model (*P* < 0.001). Although LMR-LRT and BLRT were also significantly different in the two-classes and three-classes models (*P* < 0.001); the lowest AIC, aBIC, and lower BIC in the four-classes model indicated that the model was more suitable.

In Figure 1, we listed the four identified latent classes of HRBs. The low-risk class (64.0%, n = 14,502) showed the lowest prevalence of all classes of HRBs as compared with the other classes, in which few students reported smoking (0.4%), suicidal behaviors (6.1%), drinking (7.7%), non-suicidal self-injuries (7.8%), screen time (10.7%), and some students reported unintentional injuries (35.4%). In contrast, the high-risk class (2.7%, n = 603) exhibited significantly increased incidence of all kinds of HRBs, among which 22.3% of students reported current smoking, 52.9% of students had suicidal behaviors, 58.6% of students reported screen time, and 76.5% of students reported current drinking. More importantly, the reporting rate of non-suicidal self-injuries in this class was 100%. In between, we also identified two moderate-risk classes described as moderate-risk class 1 (4.5%, n = 1012) and moderate-risk class 2 (28.8%, n = 6511). Among the students in the moderate-risk class 1, many students reported smoking (20.7%), drinking (72.2%), screen time (49.1%), but with relatively low prevalence of unintentional injuries (37.8%), suicidal behaviors (20%) and non-suicidal self-injuries (18.9%), we thus defined this class as “smoking, drinking and screen time” class. Whereas, students in moderate-risk class 2 showed a relatively low rate of smoking (1.5%), drinking (16.9%), and screen time (16.0%), but with a high incidence of non-suicidal self-injuries (86.4%), suicidal behaviors (30.4%), and unintentional injuries (72.5%), accordingly, we defined this class as “non-suicidal self-injuries, suicidal behaviors and unintentional injuries” class.

### 3.3. Multiple Logistic Regression Analysis

Table 3 shows the association between the above-described physical disorders and the latent class of HRBs. After adjusting for sociodemographic characteristics (gender, grade, registered residence, parents’ education level, any siblings, boarding on school, self-reported family economy, and number of friends) as compared with the low-risk class, all other groups were seen to be positively related to diarrhea, with OR (95% CI) of 1.682 (1.460~1.938), 1.643 (1.535~1.758), and 2.655 (2.242~3.44) for moderate-risk class 1, moderate-risk class 2, and high-risk class, respectively (*P* < 0.001 for each group vs. low-risk class). Similarly, all other groups were seen to be positively related to fever, with OR (95% CI) of 1.870 (1.595~2.192), 1.447 (1.337~1.567), and 3.135 (2.619~3.754) for moderate-risk class 1, moderate-risk class 2, and high-risk class, respectively (*P* < 0.001 for each group vs. low-risk class). All other groups were found to be positively related to cough, with OR (95% CI) of 1.379 (1.212~1.569), 1.871 (1.762~1.987), and 2.158 (1.819~2.560) for moderate-risk class 1, moderate-risk class 2, and high-risk class, respectively (*P* < 0.001 for each group vs. low-risk class), and they were also positively related to vomiting with OR (95% CI) of 2.884 (2.445~3.401), 1.748 (1.592~1.918), and 3.776 (3.113~4.581) for moderate-risk class 1, moderate-risk class 2, and high-risk class, respectively (*P* < 0.001 for each group vs. low-risk class).

## 4. Discussion

For physical disorders of adolescents, we mainly investigated the occurrence of diarrhea, fever, cough, and vomiting. Our results showed that more than one fifth (23.5%) of Chinese adolescents have diarrhea, which is higher than the results from the children aged 8 to 12 in the American Pediatric Hospital. (10.5%) [41]; we speculated that the possible reason is that adolescents are more likely to engage in HRBs (such as smoking, drinking and drug abuse, etc.) than children, which is consistent with the study of Staff, J. [42]. However, the prevalence of diarrhea in this study was close to that of the previous study (24.8%) [43]. Regarding fever in adolescents, there were very few reports, however, we found that the prevalence in Chinese adolescent was 10.7%, which is lower than a study in the children of Southeastern Ghana showing a rate of 15% [44]; these differences probably partly due to the difference in the age of the participants, namely children were more likely to develop pneumonia, which could cause fever [45]. In this study, the prevalence of cough was found to be similar to other studies (50.6%), such as that reported by Koskela, H.O.’ et al. in Finland (50.4%) [46]. In addition, our results showed that the prevalence of vomiting was 10.7%, which was lower than the results of Weaver, E.R. (12%). This could be due to the fact that adults pay more attention to hygiene, for example, washing hands before meals, which can reduce the incidence of vomiting as compared with children and adolescents who have similar HRBs [47].

This study was designed to explore the association between latent class of HRBs and physical disorders in Chinese adolescents. Four classes of HRBs were identified by LCA. These results showed that HRBs were clustered in Chinese adolescents. Moderate-risk class 1 includes smoking, drinking, and screen time. Many studies have indicated the link between smoking and drinking [48], which together cause serious damage to adolescent health [49]. Many previous studies have shown that screen time is independently associated with health risk behaviors such as smoking, etc. [50,51]. However other studies have indicated that smoking and drinking were the most common risk behavior clusters among adults [52]. Our current study suggested that this clustering phenomenon also can occur in adolescents, and that people with a high prevalence of smoking and drinking can also have long screen time. Thus, it is reasonable to find that these behaviors were clustered together. Moderate-risk class 2 includes non-suicidal self-injuries, suicidal behaviors, and unintentional injuries. There is a clear correlation between non-suicidal self-injuries and suicidal behaviors, and people with non-suicidal self-injuries are prone to have suicidal behaviors. In addition, according to Sorenson suicidal behaviors ’s research, suicidal behaviors and unintentional injuries can be included in one class [53]. Khalsa, H.M. believed that people who have suicidal behaviors can have mania or emotional disorders, which can easily lead to unintentional injuries behavior [54]. Our present study verified this relationship. Therefore, it is necessary emphasis the development of health strategies. For example, when many students engage in dangerous behaviors such as smoking, they are likely to have dangerous behaviors such as drinking and ST. Simply changing their smoking behavior could be less effective for preventing the development of physical disorders, whereas intervention on multiple behaviors could be more effective.

More importantly, in this study, we clearly showed that there is a correlation between the classes of HRBs and the above-described physical disorders. HRBs potentially cause many physical disorders in adolescents, and an increase of HRBs significantly increases the possibility of physical disorders. Compared with the low-risk group, the prevalence of physical disorders increased significantly in the moderate-risk class 1 (smoking, drinking, and screen time), which is consistent with the findings of Uddin, R. in 89 countries around the world [55]. In addition, our research also found that the moderate-risk class 2 (non-suicidal self-injuries, suicidal behaviors, and unintentional injuries) group also increased the risk of physical disorders, and some related studies have also proved this [30,56,57]. Interestingly, compared with the moderate-risk class 1, the risk of cough is higher in the moderate-risk class 2, we speculated that it could be the consequence of unintentional injuries, as studies have shown that unintentional injuries (such as asphyxia and drowning, etc.) can also cause cough [58,59]. Taken together, compared to low-risk students that had less HRBs, students with different HRBs showed significantly higher occurrence of various physical disorders, and the most significant correlation was observed in the high-risk students that had many more HRBs (Table 3). In addition, in this study, we also found students of different grades showed significantly different health conditions, which suggested that factors of age are also involved in the occurrence of physical disorders of adolescents, similar to a previous report [60].

To the best of our knowledge, there has been no systematic study to investigate the link between the classes of HRBs and physical disorders (i.e., diarrhea, fever, cough, and vomiting). This study is the first to use LCA to explore this relationship, by which we can further understand the impact of different HRBs on adolescent physical disorders. We found HRBs are positively associated with common physical disorder of adolescents. When we further analyzed the clusters of HRBs and their association with different physical disorders, we found that in the two categories of moderate risk, namely model risk class 1 (smoking, drinking, and screen time) and moderate-risk class 2 (non-suicidal self-injuries, suicidal behaviors, and unintentional injuries), the OR values of the two classes did not change much in the adolescents with diarrhea; however, regarding fever and vomiting, the OR value of the moderate-risk class 2 was reduced, but it increased in adolescents with cough (Table 3). These findings indicated that different HRBs had different effects on different physical disorders of adolescents. We speculate that the moderate-risk class 2 increased the incidence of cough due to the existence of unintentional injuries [58,59], and drinking and smoking in the moderate-risk class 1 often cause vomiting and fever [16,17].

In summary, the present study showed that there was a positive correlation between HRBs and adolescent physical disorders, i.e., more HRBs probably lead to more disorders or diseases. More importantly, we identified four classes of HRBs using LCA and further demonstrated that some classes of the HRBs cluster are significantly associated with increased risk of physical disorders in adolescents. These findings would be helpful to develop targeted intervention program for HRBs to improve the health conditions of adolescents. For example, model risk class 1 showed the highest incidence of vomiting, and therefore for the students who frequently experienced vomiting, we should consider the risk behaviors such as smoking, drinking, and screen time when formulating a health strategy, while focusing on other risk behaviors would not play a significant role. At the same time, we should also pay attention to the heterogeneity of HRBs, and the multiplicity of HRBs should also be considered for the design of intervention program.

## 5. Limitations

It should be noted that this study was conducted using a questionnaire survey, which the recall bias and reporting bias of the subjects could not be avoided, and our data was collected four years ago, with a certain degree of bias from the current situation. Next, our survey did not include all types of HRBs, which could affect the results to some extent. Moreover, our findings were based on school students and adolescents who do not attend school were not included in the report who may be more likely to report risk behaviors. In addition, this was a cross-sectional study, and therefore there was no causal relationship between the results of the study, but only suggesting the possible correlation. Therefore, further studies are warranted to overcome these limitations.

## 6. Conclusions

Health risk behaviors include certain forms of behavior, such as substance use, physical inactivity, non-suicidal self-injuries, etc., which are all related to health or disease. In this study, by focusing on the HRBs of smoking, drinking, screen time, non-suicidal self-injuries, suicidal behaviors, and unintentional injuries. We found four HRBs subgroups with associations among the physical disorders and the subgroups of HRBs, and the strength of the associations was different. The high-risk class was most associated with physical disorders. Therefore, it is possible to focus and target different HRBs when developing an intervention plan to reduce the incidence of physical disorders. 

## Figures and Tables

**Figure 1 ijerph-17-02139-f001:**
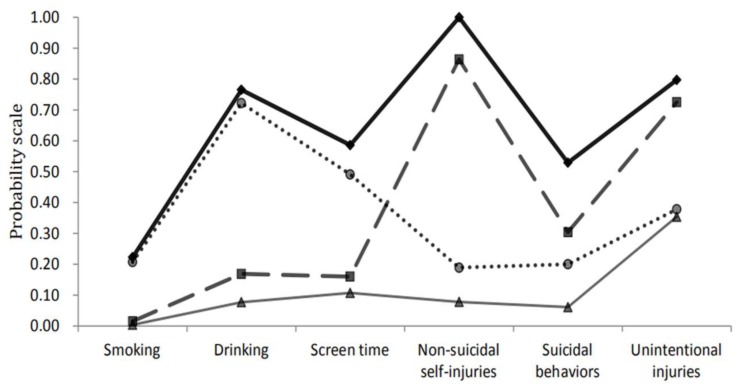
Four classes of health risk behaviors (HRBs) of the best-fitting four-class pattern. ▲ Low-risk class, 64.0%; ● Moderate-risk class 1 (smoking, drinking, and screen time), 4.5%; ◼ Moderate-risk class 2 (non-suicidal self-injuries, suicidal behaviors, and unintentional injuries), 28.8%; ♦ High-risk class, 2.7%.

**Table 1 ijerph-17-02139-t001:** Participant characteristics in the current study.

Variable	Total (*n* = 22,628)	Diarrhea		Fever		Cough		Vomiting	
		*n (%)*	*χ^2^*	*n (%)*	*χ^2^*	*n (%)*	*χ^2^*	*n (%)*	*χ^2^*
Grade			47.132 **		86.194 **		213.329 **		6.858 **
Middle school	11,993	2599 (21.7)		2168 (18.1)		6620 (55.2)		1345 (11.2)	
High school	10,635	2717 (25.5)		1441 (13.5)		4836 (45.5)		1078 (10.1)	
Gender			0.482		5.955 *		15.706 **		0.124
Male	10,990	2604 (23.7)		1820 (16.6)		5415 (49.3)		1185 (10.8)	
Female	11,638	2712 (23.3)		1789 (15.4)		6041 (51.9)		1238 (10.6)	
Registered residence			14.299 **		0.117		0.013		0.488
Rural	10,882	2267 (24.6)		1745 (16.0)		5505 (50.6)		1149 (10.6)	
Urban	11,746	2639 (22.5)		1864 (15.9)		5951 (50.7)		1274 (10.8)	
Any siblings			0.935		1.401		22.861 **		0.668
Yes	9720	2253 (23.2)		1518 (15.6)		5099 (52.5)		1022 (10.5)	
No	12,908	3063 (23.7)		2091 (16.2)		6357 (49.2)		1401 (10.9)	
Boarding on school days			41.975 **		16.089 **		2.221		14.691 **
Yes	11,320	2866 (25.3)		1695 (15.0)		5675 (50.1)		1123 (9.9)	
No	11,308	2450 (21.7)		1914 (16.9)		5781 (51.1)		1300 (11.5)	
Father’s educational level ^a^			2.439		1.580		1.163		0.012
<High school degree	13,006	3096 (23.8)		2028 (15.6)		6542 (50.3)		1381 (10.6)	
≥High school degree	9424	2159 (22.9)		1528 (16.2)		4809 (51.0)		1005 (10.7)	
Mother’s educational level ^b^			1.410		4.432 *		0.480		0.558
<High school degree	14,335	3397 (23.7)		2222 (15.5)		7232 (50.4)		1507 (10.5)	
≥High school degree	8105	1864 (23.0)		1343 (16.6)		4128 (50.9)		878 (10.8)	
Self-reported family economy			28.549 **		18.598 **		0.286		12.501 **
Bad	3240	880 (27.2)		549 (16.9)		1628 (50.2)		380 (11.7)	
General	16,345	3729 (22.8)		2506 (15.3)		8292 (50.7)		1677 (10.3)	
Good	3043	707 (23.2)		554 (18.2)		1536 (50.5)		366 (12.0)	
Number of friends			19.061 **		19.312 **		5.738		25.362 **
≤ 2	5514	1398 (25.4)		947 (17.2)		2842 (51.5)		691 (12.5)	
3–5	9620	2264 (23.5)		1417 (14.7)		4901 (50.9)		974 (10.1)	
≥ 6	7494	1654 (22.1)		1245 (16.6)		3713 (49.5)		758 (10.1)	

Note: *χ**^2^*, Chi-square test; ^a^, 198 students have no information about their father; b, 188 students have no information about their mother; *, *P* < 0.05; and **, *P* < 0.01.

**Table 2 ijerph-17-02139-t002:** Model fit statistics for each of the fitted latent class analysis models.

Statistic	2 Classes	3 Classes	4 Classes	5 Classes
AIC	120,896.912	119,991.261	119,844.588	119,822.264
BIC	121,001.263	120,151.800	120,061.315	120,095.180
aBIC	120,959.949	120,088.241	119,975.510	119,987.129
LMR-LRT	<0.001	<0.001	<0.001	0.0592
BLRT	<0.001	<0.001	<0.001	<0.001
Entropy	0.549	0.725	0.692	0.579

Note: AIC, Akaike information criteria; BIC, Bayesian information criteria; aBIC, Adjusted Bayesian information criteria; LMR-LRT, Lo–Mendell–Rubin likelihood ratio; BLRT, bootstrapped likelihood ratio test.

**Table 3 ijerph-17-02139-t003:** Association of physical disorders with different latent classes of HRBs.

Health Risk Behaviors (*n*)	Diarrhea			Fever			Cough			Vomiting		
	*n (%)*	Crude OR (95% CI)	Adjusted OR (95% CI)^a^	*n (%)*	Crude OR (95% CI)	Adjusted OR (95% CI)^a^	*n (%)*	Crude OR (95% CI)	Adjusted OR (95% CI)^a^	*n (%)*	Crude OR (95% CI)	Adjusted OR (95% CI)^a^
Low-risk class (14502)	2891 (19.9)	Ref.	Ref.	1954 (13.5)	Ref.	Ref.	6585 (45.4)	Ref.	Ref.	1185 (8.2)	Ref.	Ref.
Moderate-risk class 1 (1012)	299 (29.5)	1.684 (1.463–1.939) ***	1.682 (1.460–1.938) ***	229 (22.6)	1.878 (1.609–2.192) ***	1.870 (1.595–2.192) ***	523 (51.7)	1.286 (1.132–1.461) ***	1.379 (1.212–1.569) ***	205 (20.3)	2.855 (2.422–3.365) ***	2.884 (2.445–3.401) ***
Moderate-risk class 2 (6511)	1890 (29.0)	1.643 (1.536–1.757) ***	1.643 (1.535–1.758) ***	1222 (18.8)	1.484 (1.372–1.605) ***	1.447 (1.337–1.567) ***	3964 (60.9)	1.871 (1.763–1.986) ***	1.871 (1.762–1.987) ***	880 (13.5)	1.756 (1.601–1.927) ***	1.748 (1.592–1.918) ***
High-risk class (603)	236 (39.1)	2.583 (2.182–3.057) ***	2.655 (2.242–3.144) ***	204 (33.8)	3.283 (2.755–3.912) ***	3.135 (2.619–3.754) ***	384 (63.7)	2.108 (1.780–2.497) ***	2.158 (1.819–2.560) ***	153 (25.4)	3.821 (3.151–4.633) ***	3.776 (3.113–4.581) ***

Note: OR is odds ratio and CI is confidence interval. *** *P* < 0.001 as compared with reference. Moderate-risk class 1 (smoking, drinking, and screen time) and moderate-risk class 2 (non-suicidal self-injuries, suicidal behaviors, and unintentional injuries). ^a^ Adjusted for gender, grade, registered residence, parents’ education level, any siblings, boarding on school, self-reported family economy, and number of friends.

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
