# Peer review of "Physical Disorders are Associated with Health Risk Behaviors in Chinese Adolescents: A Latent Class Analysis"

_ijerph, 2020, doi:10.3390/ijerph17062139_

Round 1

Reviewer 1 Report

The title is too long.

If the dataw correspond to 2015-2016, why is the manuscript not submitted until 2020?

Some of the references used in the introduction section have been updated, to what extent is this theoretical foundation representative of some data that were collected 5 years ago in time?

Research hypotheses should be clearly formulated, and located at the end of the introduction section. Finally, these should be discussed in the Discussion section.

The description of the characteristics of the sample should be in the Method section, not Results.

The excessive and continued use of abbreviations ebn ek text, hinders fluent reading and comprehension. It is exhausting for the reader, since he has to continually consult the meaning of each abbreviation he finds.

Figure 1 could be clearer. Using different markers is not enough to differentiate lines if they cross each other and are also the same color.

Comments in the text referring to each figure / table are scarce, especially those in table 3.

In the Discussion section, it is necessary to comment on the hypothesis contrast.

The investigation has many limitations that have not been expressed in the corresponding section.

The conclusions should not repeat the same idea that creeps in from the introduction. Once the results have been presented and discussed with respect to the hypotheses and in relation to other works present in the literature, then with the Conclusions section, it is time to specify!

Finally, a doubt: the financing is from a call for projects of the year 2019, but ... the data collection is from 2015-2016?

Reviewer 2 Report

The manuscript aimed to clarify the potential subgroups of Health Risk Behaviors (HRBs, including smoking, drinking, screen time, non-suicidal self-injury, suicidal behaviors and unintentional injurious), and to examine the association between the subgroups of HRBs and physical disorders (diarrhea, fever, cough and vomiting). The sample included 22628 middle and high school students in six cities of China. The latent class analysis was used to define subgroups of HRBs, and logistic regression was used to test the association between HRBs and the outcomes. The study addressed a relevant research topic and it meets the scope of the journal. The background is really good and the relevance of the manuscript is based on a few studies that tested the clustering of HRBs in young people; the probable scenario in youth. However, improvements in the final report are needed before the publication. Specifically, information about the sampling procedures and the reliability of the HRB questionnaires should be included. Also, the discussion should be improved in order to address relevant topics of the results and their implications for practice and research in public health.

Abstract

  1. Considering that a large number of behaviors can be classified as “health-risk behaviors” (HBRs), please clarify which HRBs were measured in the study before presenting the results.
  2. Please specify that distinct analyses were used to combine HBRs (LCA) and to test association with physical disorders (binary logistic regression).
  3. Please explain abbreviations in the abstract (e.g., OR and 95% CI).

Method:

  1. The strategy of school selection was described. However, information about the population of the study was not reported, and this is relevant to show the external validity of the study. I suggest that authors add information on the total of eligible schools and students in each category: middle schools and high schools in rural areas, middle schools and high schools in the urban areas.
  2. Please clarify the cities evaluated in the study.
  3. Please clarify whether all students from the selected classes were invited to the study.
  4. Which were the grades invited to the study? Please clarify it.
  5. Lines 121-135: Please add information on the parameters of the reliability and validity of the questions used in the study.
  6. Lines131-135: Please clarify the scale of screen time questions.
  7. Lines 144-150: Please add information on the parameters of reliability of the questions used for suicidal behavior measurement.
  8. Please add a detailed description of the descriptive analyses, bivariate tests (e.g., Chi-square test) and the association analyses that were used in the study (bivariate and multiple logistic regression analyses).

Discussion:

  1. Lines 250-253: “Actually it is well known that the “four major causes” of morbidity and mortality in human include smoking, drinking, unhealthy diet and lack of exercise [60]. Moreover, there are also studies showing the interaction between exercise and ST [61]. Thus it is reasonable to find out these behaviors were clustered together.” I think it is not suitable to speculate that the latent class with smoking/drinking/screen time is explained by the relationship between screen time and exercise/physical activity. Physical activity and screen time have different correlates and they impact adolescent’s health by different mechanisms. Moreover, studies have shown that screen time is independently associated with other risk behaviors and behavior problems (http://dx.doi.org/10.1136/bmjopen-2018-023191), including smoking (https://doi.org/10.1111/add.13418). I suggest that authors adjust the discussion in order to meet the evaluated HRB (screen time) and why/how it may interact with other HRB (i.e., smoking/drinking) in youth.
  2. Lines 278-283: the sentences reported that the magnitude of the association between the clustering of HRBs and physical disorders may be different according to the outcome. However, no discussion on this topic was presented. Which are the reasons for this result?
  3. Please add information on the limitations of the study in the discussion.
  4. I suggest the authors add recommendations for health promotion practice based on their results. Adding examples in the discussion that are based on the results can be helpful. Also, suggestions for future researches may also be reported, considering that few studies have addressed this topic using a clustering approach.

Conclusion

  1. Please specify the HRBs that were evaluated in the study.

References

  1. The manuscript included 69 references. Most of them were cited only once or were not included in the discussion of the results. I think the authors may reduce the number of references, focusing on the primary studies of the background and theoretical support of the scope/results.

Round 2

Reviewer 1 Report

Good job!